# The Relationship between Coronary Artery Wall Shear Strain and Plaque Morphology: A Systematic Review and Meta-Analysis

**DOI:** 10.3390/diagnostics10020091

**Published:** 2020-02-08

**Authors:** Artan Bajraktari, Ibadete Bytyçi, Michael Y. Henein

**Affiliations:** 1Institute of Public Health and Clinical Medicine, Umeå University, 90187 Umeå, Sweden; artan.bajraktari@umu.se (A.B.); i.bytyci@hotmail.com (I.B.); 2Clinic of Cardiology, University Clinical Centre of Kosovo, Prishtina 10000, Kosovo; 3Institute of Environment & Health and Societies, Brunel University, Middlesex UB8 3PH, UK; 4Molecular and Clinic Research Institute, St George University, London SW17 0RE, UK

**Keywords:** wall shear strain, coronary artery disease, intravascular ultrasound

## Abstract

Background and Aim: Arterial wall shear strain (WSS) has been proposed to impact the features of atherosclerotic plaques. The aim of this meta-analysis was to assess the impact of different types of WSS on plaque features in coronary artery disease (CAD). Methods: We systematically searched PubMed-Medline, EMBASE, Scopus, Google Scholar, and the Cochrane Central Registry, from 1989 up to January 2020 and selected clinical trials and observational studies which assessed the relationship between WSS, measured by intravascular ultrasound (IVUS), and plaque morphology in patients with CAD. Results: In four studies, a total of 72 patients with 13,098 coronary artery segments were recruited, with mean age 57.5 ± 9.5 years. The pooled analysis showed that low WSS was associated with larger baseline lumen area (WMD 2.55 [1.34 to 3.76, *p* < 0.001]), smaller plaque area (WMD −1.16 [−1.84 to −0.49, *p* = 0.0007]), lower plaque burden (WMD −12.7 [−21.4 to −4.01, *p* = 0.04]), and lower necrotic core area (WMD −0.32 [−0.78 to 0.14, *p* = 0.04]). Low WSS also had smaller fibrous area (WMD −0.79 [−1.88 to 0.30, *p* = 0.02]) and smaller fibro-fatty area (WMD −0.22 [−0.57 to 0.13, *p* = 0.02]), compared with high WSS, but the dense calcium score was similar between the two groups (WMD −0.17 [−0.47 to 0.13, *p* = 0.26]). No differences were found between intermediate and high WSS. Conclusions: High WSS is associated with signs of plaque instability such as higher necrotic core, higher calcium score, and higher plaque burden compared with low WSS. These findings highlight the role of IVUS in assessing plaque vulnerability.

## 1. Introduction

Atherosclerosis is the major cause of cardiovascular (CV) disease, and related morbidity, hospitalization, and mortality worldwide [1,2]. The disease starts with increased intima-medial thickness before plaque formation and luminal narrowing. Despite the well-established stages of atherosclerosis, factors impacting plaque formation and progression in the same or different arterial segments remain not fully ascertained. Endothelial dysfunction is the earliest manifestation of atherosclerosis, followed by fatty streak formation, which is contributed to by the interplay of conventional CV risk factors, vascular biology, and local hemodynamic forces [3,4]. An important mechanical factor in the process of atherosclerosis is wall shear stress (WSS) which is the fractional force of blood exerted tangential to the vessel wall. It reflects the parallel hemodynamic force created within the endothelium of the arterial wall. WSS is determined by vascular geometry, blood properties, flow rate, and near-wall velocities [5]. Studies have shown that WSS has different effects on plaque burden and composition [6], with high WSS associated with increased plaque vulnerability [7,8,9]. However, there is no consensus on the importance and applicability of WSS in clinical practice to justify implementing its assessment routinely.

The aim of this meta-analysis is to assess the impact of different severities of WSS measured by intracardiac ultrasound (IVUS) on plaque features in coronary artery disease (CAD).

## 2. Methods

We followed the guidelines of the preferred reporting items for systematic reviews and meta-analysis (PRISMA) statement [10] amendment to the quality of reporting of meta-analyses (QUOROM) statement [11]. Due to the nature of the study design (meta-analysis), neither Institutional Ethics Review Board (IRB) approval nor patient informed consent was needed.

### 2.1. Data Sources

We systematically searched PubMed-Medline, EMBASE, Scopus, Google Scholar, the Cochrane Central Registry of Controlled Trials, and ClinicalTrial.gov, up to January 2020, using the following keywords: “Wall shear strain” OR “WSS” OR “High wall shear strain” OR “High WSS” OR “Intermediate wall shear strain” OR “Intermediate WSS” OR “Low wall shear strain” OR “Low WSS” AND “Coronary artery disease” OR “CAD” OR “Ischemic heart disease” OR “IHD” AND “Atherosclerotic plaque” OR “Plaque morphology”.

Additional searches for potential trials that included the references of review articles and the abstracts from selected congresses: scientific sessions of the European Society of Cardiology (ESC), the American Heart Association (AHA), American College of Cardiology (ACC), and European Association of Cardiovascular Imaging (EACVI) were undertaken. The wild-card term ‘‘*’’ was used to increase the sensitivity of the search strategy. The literature search was limited to studies in humans and articles published in English. No filters were applied. Two reviewers (A.B. and I.B.) independently evaluated each article. Disagreements were resolved by discussion with a third party (M.Y.H).

### 2.2. Study Selection

The criteria for inclusion in the meta-analysis were (i) studies investigating patients undergoing IVUS, (ii) reporting coronary WSS and plaque morphology, (iii) reporting types of WSS, and (iv) articles enrolling human population. Exclusion criteria were: (i) non-coronary WSS, (ii) insufficient statistical data for effect size, (iii) studies not in humans, (iv) children population, and (v) articles not published in English. Different types of WSS were defined based on magnitude values expressed with unit of dynes/cm^2^, as: low (<10 dynes/cm^2^), intermediate (≥10–25 dynes/cm^2^), and high WSS (>25 dynes/cm^2^) [12].

### 2.3. Outcome Variables

Key clinical endpoints were the relationship between coronary plaque morphology and types of WSS. Main outcome measures were coronary plaque morphology: baseline lumen area, plaque area, necrotic core area, dense calcium area, fibrous area, and fibro-fatty area.

### 2.4. Data Extraction

Eligible studies were reviewed and the following data were abstracted: (1) first author’s name; (2) year of publication; (3) study design; (4) types (severity) of WSS (high WSS, intermediate WSS, and low WSS); (5) patient demographic characteristics; (6) age and gender of study participants; and (7) IVUS measurements including: lumen area, plaque area, necrotic core area, dense calcium area, fibrous area, and fibro-fatty area, in different types of WSS.

### 2.5. Quality Assessment

Assessment of risk of bias in the studies included in the analysis was evaluated by the same investigators for each study and was performed systematically using the Quality Assessment of Diagnostic Accuracy Studies questionnaire (QUADAS-2) optimized to our study questions (Appendix A) [13]. The QUADAS-2 tool has four domains for risk of bias: patient selection, index test, reference test, and flow and timing, and three domains for applicability: patient selection, index, and reference test domains. 

### 2.6. Statistical Analysis

The meta-analysis was conducted applying the conventional statistical analysis models using the RevMan (Review Manager [RevMan] Version 5.1, The Cochrane Collaboration, Copenhagen, Denmark), and two-tailed *p* value <0.05 was considered significant. The number of patients, means, and standard deviations were pooled to weighted mean difference (WMD) and a 95% confidence interval (CI). Baseline characteristics are reported in median and range. Mean and standard deviation (SD) values were estimated using the method described by Hozo et al. [14]. Analysis is presented in forest plots, the standard way for illustrating the results of individual studies and meta-analysis. Meta-analyses were performed with a fixed-effects model and a random effect was used if heterogeneity was encountered. Heterogeneity between studies was assessed using Cochrane Q test and I^2^ index, as a guide, I^2^ < 25% indicated low, 25–50% moderate, and >50% high heterogeneity [15]. To assess the additive (between-study) component of variance, the reduced maximum-likelihood method (tau^2^) took into account the occurrence of residual heterogeneity [16]. Publication bias was assessed using visual inspections of funnel plots and Egger’s test.

## 3. Results

### 3.1. Search Results and Trial Flow

Of 2122 articles identified in the initial searches, 126 studies were screened as potentially relevant. After excluding 102 studies on the basis of title/abstract as not relevant, unrelated to study object, animal studies, review articles, letter to editor, or not in English language, the remaining 24 full-text articles were considered for inclusion in the meta-analysis. After careful assessment, 20 of the 24 articles were further excluded according to the eligibility criteria (Table 1) leaving the remaining four articles to be included in the analysis [7,9,17,18] (Figure 1).

### 3.2. Characteristics of Included Studies

Four studies (four observational) with 72 patients and 13,098 segment measurements were finally included in the analysis. The mean age of the included patients was 57.5 ± 9.5 years (68% male), of whom 69% had arterial hypertension and 25% were diabetics (Table 2).

### 3.3. Characteristics of Coronary Plaques

#### 3.3.1. Differences between Low WSS and High WSS

The pooled analysis showed that the presence of low WSS was associated with larger baseline lumen area (WMD 2.55 [1.34 to 3.76, *p* < 0.001]), smaller plaque area (WMD −1.16 [−1.84 to −0.49, *p* = 0.0007]), lower plaque burden (WMD −12.7 [−21.4 to −4.01, *p* = 0.04]) and lower necrotic core area (WMD −0.32 [−0.78 to 0.14, *p* = 0.04], Figure 2), compared to high WSS. The presence of low WSS was also associated with smaller fibrous area (WMD −0.79 [−1.88 to 0.30, *p* = 0.02]) and smaller fibro-fatty area (WMD −0.22 [−0.57 to 0.13, *p* = 0.02], Figure 3), compared to high WSS. Dense calcium score did no differ between the two groups (WMD −0.17 [−0.47 to 0.13, *p* = 0.26]).

#### 3.3.2. Differences between Intermediate WSS and High WSS

There was a smaller plaque burden in intermediate WSS compared to high WSS (WMD 9.65 [6.52 to 12.7, *p* < 0.001]). All the other coronary plaque characteristics did not differ between these two types of WSS: lumen area WMD =1.61 [−3.21 to −0.01, *p* = 0.05], plaque area, WMD 0.66 [−0.14 to 1.46, *p* = 0.11], plaque burden, WMD 9.65 [6.52 to 12.7, *p* < 0.001], and necrotic core area, WMD 0.14 [−0.06 to 0.33, *p* = 0.17, Appendix A]. Moreover, no difference was found between the two groups with respect to dense calcium score, WMD 0.06 [−0.08 to 0.20, *p* = 0.40], fibrous area, WMD 0.43 [−0.11 to 0.98, *p* = 0.12], and fibro-fatty area, WMD 0.13 [−0.09 to 0.36, *p* = 0.24, Appendix A].

#### 3.3.3. Differences between Low WSS and Intermediate WSS

No significant difference was found between low WSS and intermediate WSS with regards to baseline coronary plaque characteristics: baseline lumen area, WMD 0.96 [−0.12 to 2.04, *p* = 0.08], plaque area, WMD −0.43 [−0.99 to 0.13, *p* = 0.13], plaque burden, WMD −3.13 [−8.71 to 2.46, *p* = 0.27], and necrotic core area, WMD −0.18 [−0.45 to 0.08, *p* = 0.18, Appendix A]. Furthermore, comparing intermediate WSS with low WSS showed similar dense calcium score, WMD −0.12 [−0.26 to 0.02, *p* = 0.09], fibrous area, WMD −0.35 [−0.89 to 0.19, *p* = 0.21], and fibro-fatty area, WMD −0.05 [−0.12 to 0.02, *p* = 0.13, Appendix A].

#### 3.3.4. Features of Plaque Vulnerability according to the Type of WSS

Compared with low WSS, the high WSS had clear features for vulnerable plaques at baseline (Figure 4): higher necrotic core area, WMD −0.32 [−0.78 to 0.14, *p* = 0.04], higher plaque burden, WMD −12.7 [−21.4 to −4.01, *p* = 0.04], and higher fibrous area, WMD −0.79 [−1.88 to 0.30, *p* = 0.02] (Figure 2 and Figure 3). There was no significant difference in the features of plaque vulnerability when comparing either low WSS or high WSS with intermediate WSS (Appendix A).

#### 3.3.5. Risk Assessment of Bias

The assessment of risk of bias and applicability concerns based on the Quality Assessment of Diagnostic Accuracy Studies questionnaire (QUADAS-2) was optimized to our study questions (Appendix A) [12]. Four domains of criteria for risk of bias and three for applicability were analyzed, and the risk of bias was assessed as ‘‘low risk,’’ ‘‘high risk,’’ or ‘‘unclear risk’’. Most studies had high or moderate level of quality and clearly defined the objectives and the main outcomes (Appendix A). QUADAS-2 analysis for bias evaluation showed all domains to have low risk of bias (<30%); expected domains of applicability such as patients’ selection and index test that had high or unclear risk of 50%, due to lack of adequate exclusion and/or patient recruitment. 

## 4. Discussion

Despite the well-established description of different stages of atherosclerosis and the role of WSS as a mechanical factor in coronary plaque formation, many debates about the impact of different types of WSS on plaque features and progression exist. Some studies have shown that the presence of a normal or increased WSS has a protective effect on the endothelial function mediated by inhibition of endothelial proliferation such as anti-inflammatory effect, prevention of apoptosis of endothelial cells, and increased expression and activity of antioxidant enzymes in endothelial cells [19]. Other studies have also shown a relationship between high WSS and plaque vulnerability [20,21].

Findings: Our analysis shows that low WSS was associated with larger baseline lumen area smaller plaque area, lower plaque burden, and lower necrotic core area. Furthermore, low WSS had a smaller fibrous area and smaller fibro-fatty area compared with high WSS. In addition, no differences, in these parameters, were found between low WSS and intermediate WSS, or between intermediate WSS and high WSS.

Data interpretation: Animal studies have shown that low WSS promotes atherosclerosis development through loss of the physiological flow-oriented alignment of the endothelial cells, proliferation of smooth muscle cells, and transmigration of macrophages, that promote oxidative stress [22,23]. Although animal studies have shown varied remodeling responses to low WSS [4,24], some prospective human studies demonstrated an association between coronary low WSS and constrictive arterial remodeling [25,26]. Furthermore, constrictive arterial remodeling in low-WSS segments has been proposed as an adaptive mechanism to normalize local WSS to a more physiological, vasculo-protective level [4]. Contrary to those findings, our meta-analysis has shown that high WSS is associated with clear features for vulnerable plaque such as: higher necrotic core area, higher dense calcium score, and higher plaque burden compared to low WSS. These findings are supported by previous studies which showed that high WSS plays an important role in maintaining vulnerable plaques and proposed it as a contributor to plaque rupture and thrombosis of advanced atherosclerotic plaques in human coronary [27,28] and carotid arteries [29,30]. In addition, a human carotid autopsy study demonstrated that high WSS segments with increased macrophage levels constitute a significant substrate for plaque rupture [31].

The concept of vulnerable plaque is more complex and is not limited only to the propensity toward thrombosis [32]. Based on measures of fibrous area, a thin cap of 23 ± 19 μm near the rupture site carries higher risk compared with a thickness of 20–65 μm [33]. Such a fact suggests the possibility of potential rupture particularly when accompanied by higher necrotic core area, higher calcification, and more macrophages within the fibrous cap [33,34]. Our findings show a fibrous cap of thickness exceeding 65 μm in the low and high WSS, but the necrotic core area and calcification were more pronounced in high WSS, which is another reason to suggest the presence of vulnerable plaques in high WSS. In summary therefore, it seems that plaque cap thickness is not the sole predictor of plaque stability, as previously thought, but rather the combination of plaque contents, degree of calcification, and WSS together determine potential vulnerability. This finding is supported by the plaque stability in extensive coronary calcification compared with mild and moderate calcification which carry significant risk of acute coronary syndrome [35].

Limitations: The limitation of this study was the lack of a randomized clinical trial and the small sample volume that would limit the strength of the findings. Another limitation is the lack of clinical follow-up data on the studied patients, which would have strengthened the issue of plaque vulnerability. Future studies should focus on the relationship between plaque morphology and cardiac events. IVUS is an expensive investigation and has limited application in intervention cardiology, hence the limited number of IVUS publications that are available for analysis. We relied on the WSS measurements as they were published, so did not have any hand in controlling the accuracy of measurements.

Clinical implications: Our findings support the important role of WSS in maintaining stable coronary arterial wall function, manifested in the form of plaque morphology, contents, and vulnerability. While WSS is currently measured only invasively by IVUS, the move towards non-invasive CT-based measurements of other arterial luminal function, e.g., FFR (fraction flow reserve), could predict a future development of similar algorithms that could accurately measure WSS in high risk patients. 

## 5. Conclusions

High WSS is associated with higher necrotic core, higher calcium score, and higher plaque burden compared with low WSS, suggesting features of potential vulnerability. These findings highlight the role of IVUS in detecting the vulnerable plaque in CAD.

## Figures and Tables

**Figure 1 diagnostics-10-00091-f001:**
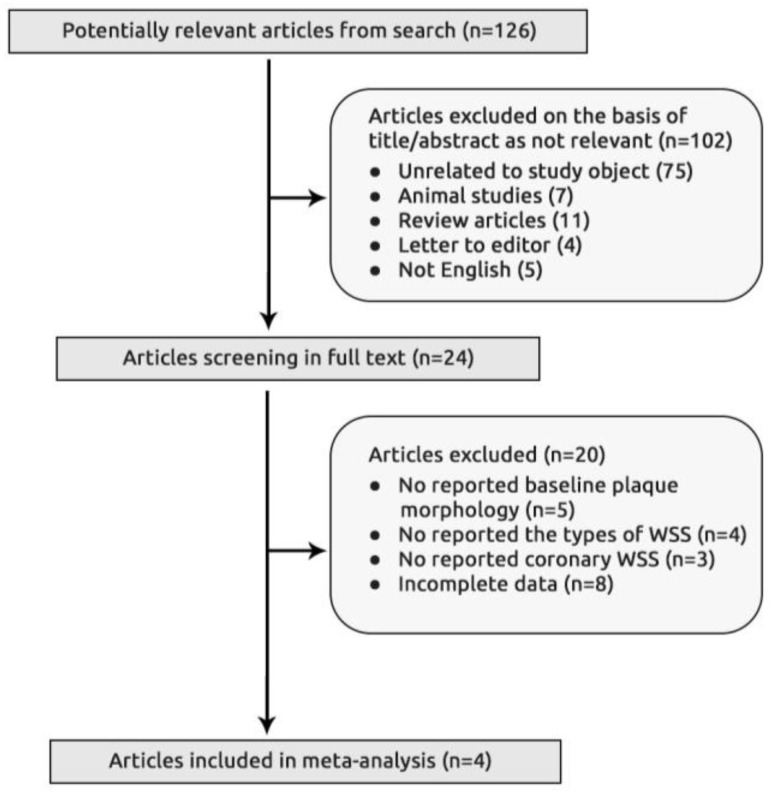
Flow chart of study section.

**Figure 2 diagnostics-10-00091-f002:**
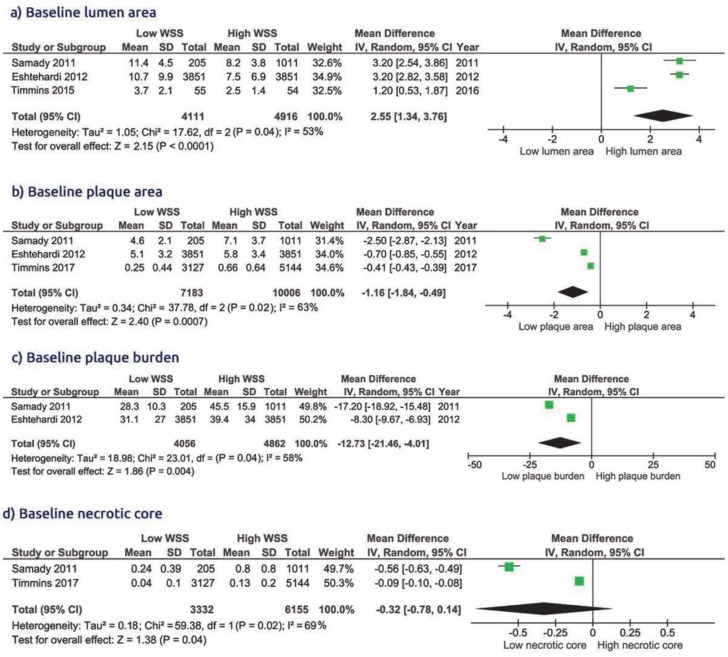
Comparison of baseline lumen area, plaque area, plaque burden, and necrotic core in group of low WSS vs. high WSS.

**Figure 3 diagnostics-10-00091-f003:**
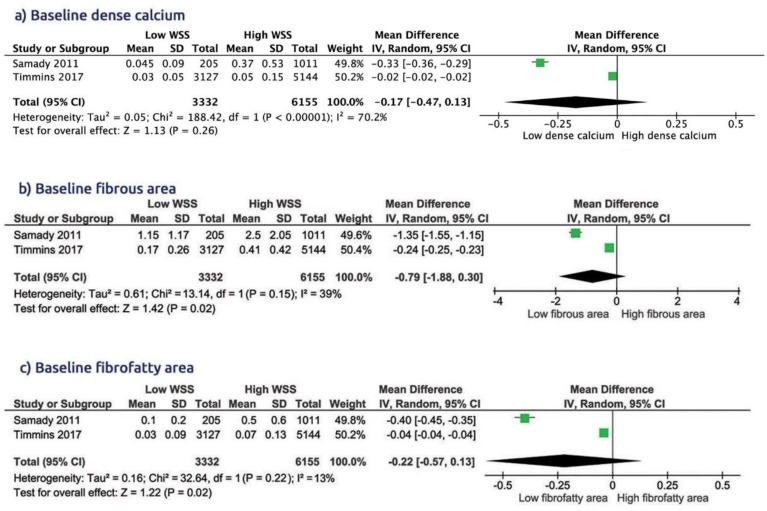
Comparison of baseline dense calcium, fibrous area, and fibro-fatty area in group of low WSS vs. high WSS.

**Figure 4 diagnostics-10-00091-f004:**
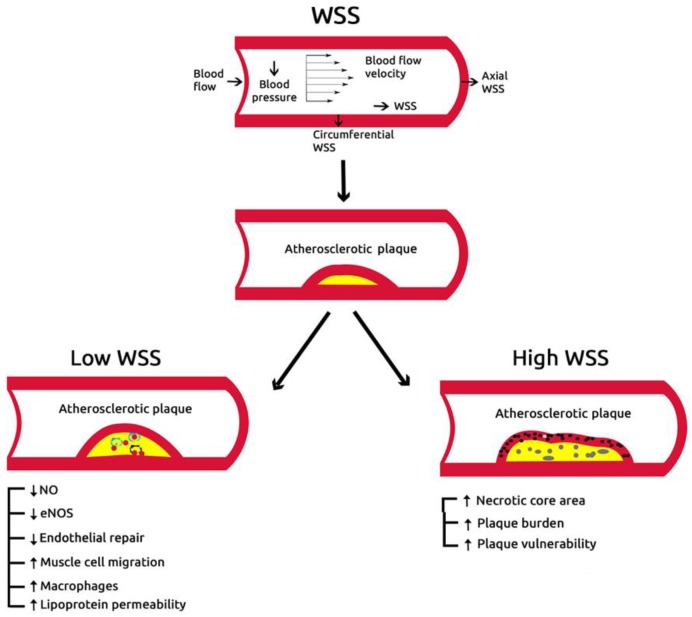
Role of wall shear stress in vulnerable plaque.

**Table 1 diagnostics-10-00091-t001:** Main characteristics of trials included in the study.

Study (Trial) Year	Study Design	Types of WSS	No. of Segments	Inclusion Criteria	Exclusion Criteria	Primary Endpoints
Samady 2011	Observational	Low WSS	2249	Abnormal	Myocardial infarction	Lumen area
	prospective	Intermediate WSS		noninvasive stress test or	Cardiogenic shock	Plaque area
	study	High WSS		stable angina	Hemodynamic instability	Necrotic core area
				syndromes	CABG or PCI	Dense calcium area
					Myocardial infarction	Fibrofatty area
					Cardiogenic shock	Fibrous area
Eshtehardi 2012	Observational	Low WSS	3581	Abnormal	Myocardial infarction	Lumen area
	prospective	Intermediate WSS		noninvasive stress test or	Cardiogenic shock	plaque area
	study	High WSS		stable angina	Hemodynamic instability	plaque burden
				syndromes	CABG or PCI	
Timmins 2015	Prospective	Low WSS	3871	CAD	NR	Lumen area
	observational	Intermediate WSS				
	study	High WSS				
Timmins 2017	Observational	Low WSS	14,235	Abnormal	NR	Plaque area
	prospective	Intermediate WSS		noninvasive stress test or		Necrotic core area
	study	High WSS		stable angina		Dense calcium area
				syndromes		Fibrofatty area
						Fibrous area

WSS: wall shear strain; CAD: coronary artery disease; CABG: coronary artery by-pass grafting; PCI: percutaneous coronary intervention; NR: non reported.

**Table 2 diagnostics-10-00091-t002:** Main characteristics of patients enrolled among trials included in the study.

Study (Trial) Year	Groups	No. of Patients	No. of Segments	Age Year	Male (%)	HTN (%)	DM (%)	TC (mg/dL)	Triglyceride (mg/dL)	Smoking (%)
Samady 2011	L-WSS	20 *	205	54 ± 10 *	65	70	35	186 ± 13	115.5+	25
	I-WSS		1034	NR	NR	NR	NR	NR	NR	NR
	H = WSS	27*	1010	NR	NR	NR	NR	NR	NR	NR
Eshtehardi 2012	L-WSS		3851 *	50 ± 10 *	60	60	26	181.5 ± 34	114 ± 95	22
	I-WSS									
	H = WSS									
Timmins 2015	L-WSS	5 *	3871 *	62.1 ± 7.6	65.5	72.2	23.6	NR	NR	NR
	I-WSS			61.7 ± 10.2	79.6	67.7	13.0	NR	NR	NR
	H = WSS			62.9 ± 10.3	74.1	74.1	16.7	NR	NR	NR
Timmins 2017	L-WSS	20 *	1785	54 ±10 *	65 *	70 *	35*	186 ± 16 *	107± 101 *	25 *
	I-WSS		413							
	H = WSS		929							

Abbreviations: L-WSS: low wall shear strain; I-WSS: intermediate wall shear strain; H-WSS: high wall shear strain; HTN: hypertension; DM: diabetes mellitus; TC: total; (*): whole group.

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
