# Peer review of "The Relationship between Coronary Artery Wall Shear Strain and Plaque Morphology: A Systematic Review and Meta-Analysis"

_diagnostics, 2020, doi:10.3390/diagnostics10020091_

Round 1

Reviewer 1 Report

Diagnostics 709011

Major concerns:

There are significant differences in the baseline lumen areas between the wall shear stress (WSS) in the Samady 2011/Eshtehardi 2012 and that in the Timmins 2015. The similar issue is also present in the baseline plaque areas. ((a) and (b) in Figure 2). The author should explain the discrepancy in between. It would also be helpful if the author can offer clear definitions for low, intermediate, and high wall shear stress, respectively.

The clinical implication of this article would be much improved if the wall shear stress could correlate with not only arterial morphology on IVUS but also cardiovascular outcomes. However, patients with major cardiovascular events, including acute myocardial infarction, unstable hemodynamics, undertaking PCI or CABG, and cardiogenic shock, were excluded in the original articles. (Timmins, et al. Interface 2017;14(127):pii:20160972. doi:10.1098/rsif. 2016.0972). This exclusion makes the clinical significance be attenuated. The author may improve this weakness by, at least in discussion, addressing the clinical significance of baseline IVUS findings and subsequent clinical outcomes.

Minor critiques:

The table 2 should be reorganized.   

Author Response

Major concerns:

1.There are significant differences in the baseline lumen areas between the wall shear stress (WSS) in the Samady 2011/Eshtehardi 2012 and that in the Timmins 2015. The similar issue is also present in the baseline plaque areas. ((a) and (b) in Figure 2). The author should explain the discrepancy in between.

Response: Thank you for suggestion. The discrepancy between these two studies in lumen area and plaque area seems to related to coronary lumen area where the plaque was localized as well as the plaque dimension. The statistical results are acceptable and the heterogeneity was moderate as well as there was no publication bias.

2. It would also be helpful if the author can offer clear definitions for low, intermediate, and high wall shear stress, respectively.

Response: Thank you for suggestion. We have added a detailed paragraph in the methods section.

3. The clinical implication of this article would be much improved if the wall shear stress could correlate with not only arterial morphology on IVUS but also cardiovascular outcomes. However, patients with major cardiovascular events, including acute myocardial infarction, unstable hemodynamics, undertaking PCI or CABG, and cardiogenic shock, were excluded in the original articles. (Timmins, et al. Interface 2017;14(127): pii:20160972. doi:10.1098/rsif. 2016.0972). This exclusion makes the clinical significance be attenuated. The author may improve this weakness by, at least in discussion, addressing the clinical significance of baseline IVUS findings and subsequent clinical outcomes.

Response: Thank you for your suggestion. Unfortunately, we do not have enough data to test the relationship between plaque morphology and cardiac events. We have added this to the limitation paragraph.

Minor critiques:

The table 2 should be reorganized.  

1.The Authors state in the Results that after the screening of the articles only 5 were included however from the Tables it is clear that the correct number is 7. Correct this inconsistency.

Response: We apologize for this mistake. We have corrected this in the revised manuscript.

Reviewer 2 Report

The paper is well written and the study design seems sound. I think this analysis represents a interesting topic for readers that has not yet been adressed to full extent, at least in form of meta-analyses. Maybe the introduction could be a little improved, e.g. adress the progression of atherosclerosis and plaque formation in more detail, endothelial dysfunction and more clinical context (acute coronary syndrome).

Author Response

1.The paper is well written and the study design seems sound. I think this analysis represents a interesting topic for readers that has not yet been addressed to full extent, at least in form of meta-analyses. May be the introduction could be a little improved, e.g. address the progression of atherosclerosis and plaque formation in more detail, endothelial dysfunction and more clinical context (acute coronary syndrome).

Response: Thank you for suggestion! we have revised the introduction along the lines of your comments.

Round 2

Reviewer 1 Report

No further comments.